# LncRNAs in the Type I Interferon Antiviral Response

**DOI:** 10.3390/ijms21176447

**Published:** 2020-09-03

**Authors:** Beatriz Suarez, Laura Prats-Mari, Juan P. Unfried, Puri Fortes

**Affiliations:** 1Program of Gene Therapy and Hepatology, Center for Applied Medical Research (CIMA), University of Navarra (UNAV), 31008 Pamplona, Spain; bsuarez.1@alumni.unav.es (B.S.); lpratsmari@alumni.unav.es (L.P.-M.); 2Navarra Institute for Health Research (IdiSNA), 31008 Pamplona, Spain; 3Centro de Investigación Biomédica en Red de Enfermedades Hepáticas y Digestivas (CIBEREHD), 28029 Madrid, Spain

**Keywords:** LncRNAs, type I IFN, antiviral response

## Abstract

The proper functioning of the immune system requires a robust control over a delicate equilibrium between an ineffective response and immune overactivation. Poor responses to viral insults may lead to chronic or overwhelming infection, whereas unrestrained activation can cause autoimmune diseases and cancer. Control over the magnitude and duration of the antiviral immune response is exerted by a finely tuned positive or negative regulation at the DNA, RNA, and protein level of members of the type I interferon (IFN) signaling pathways and on the expression and activity of antiviral and proinflammatory factors. As summarized in this review, committed research during the last decade has shown that several of these processes are exquisitely regulated by long non-coding RNAs (lncRNAs), transcripts with poor coding capacity, but highly versatile functions. After infection, viruses, and the antiviral response they trigger, deregulate the expression of a subset of specific lncRNAs that function to promote or repress viral replication by inactivating or potentiating the antiviral response, respectively. These IFN-related lncRNAs are also highly tissue- and cell-type-specific, rendering them as promising biomarkers or therapeutic candidates to modulate specific stages of the antiviral immune response with fewer adverse effects.

## 1. LncRNAs

It is estimated that more than 70% of the human genome is transcribed into RNA; however, only about 1% encodes for proteins [1]. Given that most coding genes are conserved and the high organismal complexity of mammals, it is not surprising that the non-coding genome has evolved varying levels of regulatory potential to play relevant roles in mammalian cells. Part of this regulation is mediated by non-coding RNAs (ncRNAs). Indeed most ncRNAs function to allow and modulate gene expression and can be classified into small ncRNAs: miRNAs (micro RNAs), siRNAs (small interfering RNAs), piRNAs (P-element-induced wimpy testis (Piwi)-interacting RNAs), snoRNAs (small nucleolar RNAs), snRNAs (small nuclear ribonucleic acid), and tRNAs, or longer RNAs that comprise most ribosomal RNAs (rRNA) and a complex family named long non-coding RNAs (lncRNAs).

LncRNAs are transcripts longer than 200 nucleotides with poor coding capacity. This broad definition helps to define a highly heterogeneous group of ncRNAs that can accomplish a remarkable variety of biological functions in the maintenance of cell physiology and the development of disease. LncRNAs share several features with their coding mRNA (messenger RNAs) counterparts. Both are transcribed by RNA Polymerase II (RNA pol II), and their gene loci show similar histone modifications associated with transcriptional initiation (histone 3 trimethylation of lysine 4 (H3K4me3)) and elongation (histone 3 trimethylation of lysine 36 (H3K36me3)). LncRNAs also undergo 5′ capping, polyA tailing, and co-transcriptional splicing, though less efficiently than mRNAs [2]. However, compared to mRNAs, lncRNAs are expressed at much lower levels and in a tissue-, cell-type-, and cell-state-specific manner [1,3,4,5]. LncRNAs are also less conserved at the primary sequence level. However, their secondary structure (especially at the 5′ end) is better evolutionarily preserved, suggestive of a non-neutral selective pressure to retain functional structural motifs [2,6]. LncRNAs accumulate preferentially in the nucleus [7]. Many nuclear lncRNAs are found tethered to chromatin or accumulated in nuclear bodies, where they play regulatory functions. Nuclear retention of lncRNAs requires the development of mechanisms to escape nuclear degradation and export to the cytoplasm. Some lncRNAs can be stabilized and retained in the nucleus thanks to the presence of snoRNAs (sno-lncRNAs) protecting their ends [8,9]. Other non-polyadenylated lncRNAs have a blunt-ended triplex RNA structure at the 3′ end to prevent decay, similar to some nuclear viral RNAs [10,11,12].

LncRNA genes are numerous, and the number of annotated non-coding genes has been increasing dramatically over the last years. Currently, there are 17,952 lncRNAs (not including pseudogenes) annotated in the last release of Gencode (GRCh38.p13), and many more are predicted to exist [1]. Such high numbers of lncRNA genes inevitably give rise to a wide variety of biotypes. They can be classified according to features such as their genomic context, their functional localization (in *cis* or *trans*), or the type of processes they regulate (transcriptional or post-transcriptional) [13].

Regarding their genomic context, lncRNA genes can be intergenic (lincRNAs), when they are transcribed autonomously from non-coding regions between coding genes, or they can be located close to or within protein-coding genes (PCGs) (Figure 1). In the latter case, they can be subclassified as (i) intronic, which are transcribed from intronic sequences of PCGs, (ii) sense, transcribed from the same strand of a PCG, (iii) antisense, transcribed from the opposite strand of a PCG, (iv) divergent, transcribed head to head (occasionally from a bidirectional promoter) in the opposite direction to the PCG, (v) in tandem, transcribed in the same orientation, but head to tail either upstream or downstream of the PCG, or (vi) convergent, transcribed tail to tail in the opposite direction to the PCG. Also, lncRNAs can be transcribed from enhancer (eRNAs) and promoter (PROMPTs) sequences [14,15].

According to their mechanism of action, lncRNAs can be *trans*-acting, when they function at distal genomic regions from their site of transcription, or *cis*-acting, when they remain co-transcriptionally tethered to their loci to regulate the expression of genes in close genomic proximity [6]. Such *cis* activity is a unique feature of lncRNAs. It allows localization and binding of the lncRNA at the target locus without the need for specific domains, as opposed to protein factors. Also, co-transcriptional retention of *cis*-acting lncRNAs allows prompt responses to stimuli, as they do not require transport between cellular compartments. Surprisingly, some *cis*-acting lncRNAs can function over distant genomic locations, not limited to the vicinity of their loci [15].

*Cis*-acting lncRNAs can only function as regulators of transcriptional processes, whereas *trans*-acting lncRNAs can also regulate post-transcriptional processes and protein activity (Figure 2). LncRNAs can control chromatin states through epigenetic modifications. They can recruit chromatin-activating complexes such as WDR5-MLL (WD (Tryptophan-Aspartate) repeat domain 5—mixed lineage leukemia complex), that lays H3K4me3 marks, or silencing complexes such as PRC2 (polycomb repressive complex 2), that lays histone 3 trimethylation of lysine 27 (H3K27me3) marks [15]. The simple act of transcription of a lncRNA can alter the epigenetic state and chromatin interactions of the genomic region, affecting the expression of PCGs located nearby [16].

LncRNAs can also affect transcription by transcriptional interference, a common event during genomic imprinting, and by affecting the activity of promoters and enhancers [6]. LncRNAs can accumulate at promoters and alter RNA pol II occupancy and the formation of the preinitiation complex (PIC), necessary for DNA unwinding and RNA pol II positioning at the TSS (Transcription Start Site) [16,17,18]. Also, lncRNAs can modify the availability of enhancers through chromatin looping, competing, or favoring the assembly of the Mediator complex. This multiprotein hub helps the recruitment of specific transcription factors (TFs) and induces transcription initiation and elongation [2,19,20]. Some eRNAs have been described to act in *cis* and play a fundamental role in enhancer regulation [15]. They are usually divergent transcripts and share a minimal promoter with their coding counterpart [21]. Interestingly, in some cases, eRNA regulatory activity is sequence-independent, and the transcription of the DNA elements within the eRNA locus may reinforce the binding of TFs [22].

At the post-transcriptional level, lncRNAs can be involved in the regulation of splicing, trafficking between cellular compartments, translation, and RNA stability [23]. LncRNAs have been described to hybridize directly with pre-mRNA sequences to block splicing or to interact with different components of the spliceosome machinery at nuclear domains like speckles, to modulate the efficiency of this process [24]. Many RNA processing-related activities can occur in membrane-less compartments, where lncRNAs act as scaffolds. For example, NEAT1 (nuclear-enriched abundant transcript 1) is indispensable for nuclear paraspeckle formation. In fact, some paraspeckle-related proteins such as NONO (non-POU domain containing octamer binding), PSPC1 (paraspeckle component 1), and PSF (polypyrimidine tract-binding protein-associated splicing factor), regulate NEAT1 transcription [25]. Translational control is exerted in the cytoplasm by interacting with the translation machinery or by altering the stability of target mRNAs [26]. LncRNAs can belong to the cytoplasmic ½-sbsRNAs (half-staufen1 binding site RNAs) group, which contains Alu repeats. These repeats bind through imperfect base-pairing to the Alu elements present in the 3′ untranslated region (3′UTR) of actively transcribed genes. The resulting secondary structure can be recognized by staufen1, which mediates the decay of target mRNAs [27]. On the contrary, lncRNAs containing SINEB2 (short interspersed nuclear element B2) repeats can bind to mRNA 5′ ends, enhancing the recruitment of ribosomes and inducing translation [28]. Under stress conditions, when cap-dependent translation is inhibited, lncRNAs can hybridize to target pre-mRNAs favoring the formation of an internal ribosomal entry site (IRES) to facilitate cap-independent translation of stress-response mRNAs [29,30]. Regulation of translation and stability can also be exerted by affecting miRNAs. LncRNAs can bind directly to the mRNA, to mask the miRNA target site, or act as competing endogenous RNAs (ceRNAs). CeRNAs are lncRNAs, pseudogene transcripts, mRNAs, or circular RNAs that contain miRNA target sites. They sequester miRNAs through base pairing, increasing the expression of the miRNA mRNA targets, as they escape from RISC (RNA-Induced Silencing Complex) complex-triggered silencing [2].

LncRNAs can also alter protein function and localization. LncRNA binding to human antigen R (HuR), a mRNA stabilizing protein, promotes the transport of HuR from the nucleus to the cytoplasm, where it stabilizes mRNAs that promote tumor initiation and progression [31,32,33,34]. LncRNAs are also involved in various post-translational modifications of proteins [35]. They act as scaffolds that favor the interaction of methylases, demethylases, kinases, phosphatases, acetylases, deacetylases, or E3 ubiquitin-ligases with their targets, or by hiding the protein region that should be modified. LncRNAs can modulate protein binding to multiple enzymes and regulators, allowing synergistic effects [36,37]. Furthermore, lncRNAs can modulate enzymatic activity by inducing conformational or allosteric changes that regulate the docking site of the target molecules [37,38,39]. Lastly, and paradoxically, some lncRNAs can harbor small open reading frames (smORF) that encode for micropeptides with less than 100 codons, some of which are conserved. This is suggestive of a biological function preserved through evolution rather than the result of spurious ribosome binding [40,41,42,43].

The regulation exerted by lncRNAs spans all fundamental steps of cell growth and differentiation, homeostasis, and response to stimuli. It has been speculated that, since lncRNAs are fast evolving molecules, mammalian or primate-specific lncRNAs could play a vital role in processes that are heavily targeted by evolutionary selection, such as viral infections and the antiviral response [44,45].

## 2. The Antiviral Interferon Response

Viral infections have shaped the genomes of prokaryotes and eukaryotes for millions of years. Endogenous retroviruses alone are estimated to have contributed to half of the sequences of the human genome that remain as relics of germline infections that became evolutionarily fixed [46]. This phenomenon required a host that was permissive to the virus while, at the same time, capable of limiting viral replication to avoid being overrun by infection. Moreover, this defense mechanism had to be self-limiting to allow the return to homeostasis after viral clearance. The combination of processes that allows a cell to balance its defense strategy against microbial insults is part of what we call the immune system. The immune system coordinates innate and adaptive responses to constitute one of the most robust defenses against damaging extracellular agents. In this review, we will focus on type I interferons (IFNs), which are cytokines with a critical role in the antiviral response.

IFNs were discovered in the late 1950s and were named after their function to “interfere” with viral replication [47,48]. There are three main types of IFNs: Type I IFN includes 13 subtypes of IFN-α, IFN-β, IFN-ε, and IFN-ω (Figure 3). The only member of type II IFN is IFN-γ, and type III members are IFN-λ1, IFN-λ2, and IFN-λ3. All of them have structural similarities with interleukin-10 (IL-10), an anti-inflammatory cytokine that plays a critical role in limiting the immune response to pathogens, ensuring tissue homeostasis [49]. Regardless of the large number of IFNs, there are only three IFN receptors, one specific to each type. Type I IFN binds to IFN alpha receptor (IFNAR), type II to IFN gamma receptor (IFNGR), and type III to IFN lambda receptor (IFNLR), each composed of two subunits: IFNAR1 and IFNAR2, IFNGR1 and IFNGR2, and IFNLR1 and IL10Rβ (interleukin 10 receptor subunit beta), respectively [50,51]. After IFN binding to the receptor, both type I and III IFNs trigger a signaling cascade that leads to the nuclear translocation of the ISGF3 complex (interferon-stimulated gene factor 3). However, while the type I receptor is ubiquitous, the type III receptor is mainly expressed in epithelial cells (Figure 3). Similarly, type II and III IFNs induce the translocation of the gamma activation factor (GAF) complex, formed by two phosphorylated STAT1 (signal transducer and activators of transcription) proteins. ISGF3 and GAF bind to the IFN-stimulated response element (ISRE) or the gamma activation sequence (GAS) respectively, ultimately leading to the transcriptional activation of IFN-stimulated genes (ISGs). Type III IFN can signal through both axes. Indeed, some ISGs are regulated by the three types of IFN, whereas others are regulated only by a specific type [51,52,53,54]. This depends on the promoter sequence of each ISG. For example, the promoters of *ISG47* and *IRF1* (interferon regulatory factor 1) contain GAS and ISRE, and all three types of IFN can stimulate their transcription. In contrast, the promoters of *2′-5′-oligoadenylate synthetase* (*OAS*), *Mx dynamin like GTPase 1* (*MxA*), *guanylate binding protein (GBP*), or *interferon regulatory factor* 7 (*IRF7*) can only respond to type I IFN [55,56].

For the antiviral function of type I IFN, first, pathogens must be sensed to trigger IFN synthesis. Then, IFN is secreted to activate IFN signaling. Type I IFN synthesis and signaling pathways can be induced in any type of nucleated cell to protect them from viral or microbial infections.

### 2.1. IFN Synthesis

Bacteria, fungi, parasites, and viruses produce molecules that are normally absent or mislocalized in a healthy cell and that are collectively called pathogen-associated molecular patterns (PAMPs). Also, microbial infections, stress, or non-programmed cell death leads to the release of factors named damage-associated molecular patterns (DAMPs). PAMPS and DAMPs are immediately detected by cell sensors known as pattern recognition receptors (PRRs) [57,58]. Four major families of PRRs have been described to date: Toll-like receptors (TLRs), retinoic acid inducible gene (RIG)-like receptors (RLRs), nucleotide oligomerization domain (NOD)-like Receptors (NLRs), and C-type lectin receptors (CLRs). They differ in their subcellular localization (plasma membrane (TLR-CLR), endosomes (TLR), or cytoplasm (RLR and NLR)), in the PAMPs that each member binds to, and in the signaling activated after PAMP recognition. Interestingly, members of different PRR families can recognize the same PAMP, and the antiviral response can be enhanced or regulated after the co-stimulation of different sensors (reviewed in reference [59]).

In the case of viruses, the major PAMPs are structural proteins from the viral capsid or envelope, which are recognized by TLR2 and TLR4 at the cell membrane, and viral DNA or RNA which is free at the extracellular environment or released into the cytoplasm of the host cell upon infection (reviewed in reference [60]). Viral DNA can be recognized in endosomes by TLR9, in the nucleus by hnRNPA2B1, and in the cytoplasm by several sensors such as absent in melanoma 2 (AIM2) or GMP-AMP synthase (cGAS). Alternatively, DNA-binding factors that participate in other cellular processes, such as polymerase III or the DNA damage response proteins Ku70 and DNA-PKcs (DNA-dependent protein kinase, catalytic subunit) can also participate in viral DNA recognition [61,62]. Similarly, linear or circular viral RNA of different sizes can be detected by TLR3, 7, and 8 (inside endosomes), NLRP3 (NACHT Domain-, Leucine-Rich Repeat-, and Pyrin-containing Protein 3), RIG-I, or MDA5 (melanoma differentiation-associated gene five receptor). Other non-canonical sensors like protein kinase regulated by RNA (PKR), DEAD-box helicase 3 (DDX3), or DEAH-box helicase 33 (DHX33) can also sense viral RNA [63].

After ligand binding, all TLRs but TLR3 use Mal (myelin and lymphocyte protein) and MyD88 (myeloid differentiation factor-88) adaptors to activate IRAK (interleukin-1 receptor-associated kinase 1) and TRAF6 (tumor necrosis factor receptor-associated factor 6). These induce MAPK (mitogen-activated protein kinase), NF-κB (nuclear factor kappa beta), and ATF2 (activating transcription factor 2) signaling, and TBK1 (TANK-binding kinase) and IRF7 transcription. TLR3 and TLR4 can use TRAM (translocation-associated membrane protein) as an adaptor to activate TRIF (TIR-domain-containing adapter-inducing interferon-β) and TBK1 and initiate IRF3-mediated transcription [64,65]. They also induce TRAF6 and RIP1 for NF-κB stimulation. Similarly, RNA recognition by RIG-I or MDA5 in the cytoplasm activates oligomerization of MAVS (mitochondrial antiviral-signaling protein) adaptor and activation of TRAF (TNFR-associated factor), TBK1, and IKK (Iκbeta kinase), leading to transcription of NF-κB, ATF2, IRF3, and IRF7 target genes. A similar pathway is induced by cGAS, which synthesizes a cyclic dinucleotide (CDN) second messenger, after the detection of DNA in the cytoplasm. CDN activates stimulator of interferon genes (STING), which induces TBK1, IRF3, and NF-κB activation [66]. Finally, cytosolic DNA sensors and NLRs may employ similar pathways or constitute the inflammasomes, which involve adaptors that link the sensor to Caspase 1. Caspase 1 is activated and regulates the processing and secretion of immature IL1b and IL18 into their active forms, leading to increased inflammation [67,68]. The outcome of these pathways is the transcription of inflammatory genes induced by ATF2 and NF-κB, the transcription of specific antiviral genes induced by IRF3 and IRF7, and the synthesis of type I IFN that results from the combined action of ATF2, NF-κB, and IRF transcription factors on the IFN promoter.

### 2.2. IFN Signaling

After synthesis, type I IFNs are released to the extracellular space where they may induce autocrine or paracrine signaling. In this manner, the antiviral response will be activated in both infected and neighboring cells, preparing them to counteract the incoming infection. First, IFN-α and -β bind with high affinity to the IFNAR2 subunit. Then, IFNAR1 is recruited, resulting in dimerization of the two chains and their auto-phosphorylation. Next, the associated janus activated kinase (JAK1), and tyrosine kinase 2 (Tyk2) are also phosphorylated. Once activated, these two proteins will phosphorylate the intracellular domains of the receptors to establish docking sites for STAT1 and STAT2, which will also be phosphorylated. Phospho-STAT1 and STAT2 form a heterodimer that binds IFN regulatory factor IRF9 in the cytoplasm, forming the ISGF3 complex [69]. ISGF3 will be translocated into the cell nucleus, where will bind the ISRE in the promoter of ISGs. Similar to type III IFN, type I IFN can also induce STAT1 homodimerization, to activate ISGs with GAS sequences, and STAT3 homodimerization, which has an anti-inflammatory action by binding to SBE (smad binding element) [52,70,71,72]. Alternative pathways include ISG activation by un-phosphorylated STAT1 or by STAT2-IRF9 heterodimer [52]. Activation of the JAK kinases also induces stress-response pathways, such as phosphatidylinositol 3-kinase (PI3K), NF-κB, and MAPK, which help to amplify the IFN response. In this manner, viral-induced stress leads to the stimulation of antiviral genes but also the induction of inflammation, proliferation, differentiation, migration, or apoptosis [73].

The genes induced by IFN signaling can be classified into primary and secondary response genes [52]. Primary IFN gene promoters have a permissive chromatin state at homeostasis, probably acquired by the action of PU.1 and other lineage-specific factors during lineage commitment. Transcription of most primary IFN genes does not require chromatin remodeling and is achieved in an early stage after IFN signaling. Their promoters are rich in CpG islands and contain high basal levels of histone 3 lysine 9 (H3K9) acetylation and preassembled polymerase II. Further acetylation at H4 recruits bromodomain-containing 4 (BRD4) and fires RNA pol II transcription elongation [74]. Instead, expression from secondary IFN genes requires the action of several regulatory complexes to acquire a permissive chromatin state that allows transcription and the translation of novel proteins, which are synthesized during the primary response.

Each ISG has one or several specific antiviral activities that, in most cases, only have a partial effect on viral replication. However, several ISGs act in coordination to achieve an efficient antiviral response [75]. ISGs can block different steps of the viral cell cycle: virus entrance, viral gene expression, genome replication, assembly, and release. Interesting examples are the Mx proteins, best known for inhibiting negative-stranded RNA viruses. By interacting with the viral ribonucleoprotein complex, MxA and MxB interfere with the intercellular trafficking of viral components, blocking the early stages of viral replication [76]. PKR is activated by dsRNA and inhibits eukaryotic initiation factor 2 (eIF2), thus stopping canonical protein translation [77]. After translation, viral proteins can be modified and inhibited by the ubiquitin-like modifier ISG15 [78]. Similarly, viral dsRNA can be altered by deamination of adenosine carried out by the adenosine deaminase acting on RNA 1 protein (ADAR1) [72], and viral DNA can be hypermutated by the apolipoprotein B editing complex (APOBEC3) [79]. Viral dsRNA can also be directly degraded through the activation of RNAse L, mediated by oligoadenylate synthase proteins (OAS1, OAS2, OAS3, OASL) [80]. Members of the family of IFN-induced proteins with tetratricopeptide repeats, IFIT1, IFIT2, IFIT3, and IFIT5, execute their antiviral functions by forming multiprotein complexes with cellular and viral proteins through their structural tetratricopeptide motifs [81].

### 2.3. IFN Regulation

Once the IFN response has been initiated, complex regulatory mechanisms control the balance between the intensity and duration of the response and the return to homeostasis. Altering the activity of the regulators breaks this delicate equilibrium and results in immune disorders ranging from fatal or chronic infections to inflammatory and autoimmune diseases. A sustained over-reaction of the IFN response can be damaging for the cell and can cause interferonopathies like systemic lupus erythematosus (SLE) or rheumatoid arthritis [82,83]. The major mechanisms of regulation required for a healthy and balanced IFN production and signaling involve transcription induction or repression and activation, inhibition, internalization, miss-localization, or degradation of protein mediators, and their binding to other regulators (reviewed in references [59,71,79,84]).

Protein mediators of the IFN pathways are subjected to tight control at the post-translational level by phosphorylation, methylation, acetylation, succinylation, and modification by ubiquitin and by ubiquitin-like moieties such as small ubiquitin-like modifier (SUMO) or ISG15 (reviewed in references [71,79]). Phosphorylation and de-phosphorylation cycles exerted by kinases and phosphatases are significant regulators of the cascades that lead to IFN synthesis and signaling. Binding of PAMPs and DAMPs to cellular sensors, and binding of IFN to its receptor, induces the phosphorylation and activation of the IFN receptor, adaptors (i.e., MAVS, STING, ASC, JAK, TYK), mediators (i.e., TRIF, TRAF, IKKs, TBK1), and transcription factors (i.e., IRFs, STATs, and NF-κB) (see above), which is then abrogated by de-phosphorylation. This only serves as a general rule, as phosphorylation of TRAF6 by the MST4 (Serine/Threonine-Protein Kinase 26) kinase prevents TRAF6 oligomerization, ubiquitination (ub), and signaling, leading to the exacerbated inflammation observed in MST4 knockdown mice [79,85]. Indeed, ub and deubiquitination (deub) are also significant regulators of the IFN pathway where linear-linked poly-ubiquitination (polyub) and ub on several lysines have been described. In general, K48-linked polyub induces proteasomal degradation while K63-linked ub is required for signaling transduction. NF-κB and IRF3 activation involve K63-linked polyub of TRAF6, TRAF3, TBK1, RIG-I, and STING, among others. Instead, K48-linked polyub marks for the proteasomal degradation of TLR3, 4 and 9, MAVS, RIG-I or STING, Mal, MyD88, TRAF1-6, and TBK1, decreasing IFN or IκB synthesis, leading to the nuclear translocation of NF-κB and increased inflammation. Transcription factors such as IRF3, IRF7, NF-κB p65 subunit, or STAT1 and STAT2 are also degraded after polyub. Ub and deub are carried out by ub-modifying complexes and de-ubiquitinating enzymes and regulated by a myriad of factors that strongly affect the antiviral response, with SOCS1 (suppressor of cytokine signaling) being the first described [86]. Similarly, SUMOylation has been shown to modify IFN mediators and influence on protein stability or activity. SUMOylation increases the activity of RIG-I and MDA5 [87,88] and suppresses the activity of p65 [89]. De-SUMOylation of IRF3 regulates the ub and degradation of IRF3 [90]. PIAS1 (protein inhibitor of activated STAT 1) is a well-known regulator of IFN that SUMOylates STAT1 [82]. Finally, the role in the antiviral response of non-conventional post-transcriptional modifications has been demonstrated during the last years. Methylation of p65 at K310 is inhibitory [91], whereas deacetylation of TBK1 and RIG-I fosters IFN production [92,93].

Post-transcriptional regulation is emerging as a novel regulator of the IFN pathways by m^6^A (N6-methyladenosine) and m^5^C (5-methylcytosine) modification of RNA targets such as MAVS, TRAFS, and SOCS3, which lead to nuclear retention or increased stability, respectively [84]. In turn, transcriptional regulation by epigenetic modifications is critical for the control of the IFN response. This involves histone modifications and DNA methylation. In fact, ISGs are hypomethylated in SLE [94]. After viral infection or IFN production, ISG promoters are controlled by histone acetylation, methylation, and monoubiquitination (reviewed in references [72,95]). Histone acetylation is critical for transcription from both primary and secondary response genes (see above). Secondary ISGs are activated by histone acetyltransferases (HATs) and inhibited by histone deacetylases (HDACs), leading to decreased response, although HDAC inhibitors can decrease the expression of proinflammatory genes [96]. Crucial negative regulation is also in charge of G9a by deposition of H3K9me2 marks, and of SETDB2 (SET domain bifurcated histone lysine methyltransferase 2), which is induced by influenza virus infection and leads to trimethylation and silencing [97,98]. Expression of IRF1 transcription factor can be silenced by enhancer of zeste homolog 2 (EZH2)-mediated deposition of H3K27me3 [99]. Interestingly, some of these signals affect transcription in a cell-type-specific manner, or they only affect individual ISG promoters [97]. This raises the relevant question of how regulation and specificity are controlled. The massive deregulation of lncRNAs observed after viral infection and IFN responses have prompted the hypothesis that lncRNAs could play fundamental roles in IFN regulation. Increasing evidence is being raised in agreement with this hypothesis [56].

## 3. LncRNAs and the IFN Response

LncRNA expression arrays and, especially, next-generation sequencing technologies have aided the discovery of the outstanding deregulation of lncRNA transcription that follows viral infections or the activation of the type I IFN pathway. Several viruses express viral lncRNAs that play critical regulatory functions in the infected cell [100,101]. Also, human and mouse viruses promote profound changes in the expression of cellular lncRNAs [102,103,104,105]. Interestingly, different viruses deregulate the same host lncRNAs in infected cells as a consequence of the activation of the antiviral response [102,104,106]. Out of the myriad of viral-induced lncRNAs described to date, only a handful have been thoroughly studied at the molecular level. An outstanding percentage of them function to affect immunity-related host factors and prevent or promote viral replication. These will be reviewed in detail.

### 3.1. General Features of Viral-Induced lncRNAs Unrelated to the Antiviral Response

A minority of virus-regulated host lncRNAs have been shown to work independently of IFN signaling (Table 1). In the frontier of what can be considered of viral or cellular origin, are cellular non-coding sequences derived from endogenous retroviruses (ERVs) such as long terminal repeats (LTRs), which can function to regulate transcription or can be expressed as lncRNAs [107,108]. This is the case of human endogenous retrovirus subfamily H (HERVH), which binds OCT4 (octamer-binding transcription factor 44), Mediator subunits, and coactivators, to allow human embryonic stem cell (hESCs) identity. True virus-regulated host lncRNAs can be divided into those that affect the function of viral proteins and those that modify cellular factors. Among some outstanding lncRNAs that regulate viral proteins are human immunodeficiency virus (HIV)-related NRON (ncRNA repressor of the nuclear factor of activated T cells) that links Tat to ubiquitin and proteasome components leading to Tat degradation and HIV latency [109]. Also the influenza virus-induced lncRNAs PAAN (PA-associated noncoding RNA) and IPAN (influenza virus PB1-associated noncoding RNA) which interact with the PA and PB1 subunits of the viral RNA dependent RNA polymerase (RdRp) complex to promote replication by helping proper complex formation (PAAN-PA) or PB1 stabilization (IPAN-PB1) [110,111]. Other lncRNAs interact with cellular factors that affect critical processes in viral replication. HIV-related HEAL (HIV-1 enhanced lncRNA) binds fused in sarcoma (FUS), helping HIV transcription [112]. HSV-related MAMDC2-AS1 binds heat shock protein 90 alpha (hsp90α) and helps in the nuclear translocation of VP16 (viral protein 16), essential to activate immediate transcription of early genes and herpes simplex virus (HSV) infection [113]. Interestingly, lncRNAs can also potentiate cell metabolism contributing to viral replication. Hepatitis C Virus (HCV)-induced HULC (hepatocellular carcinoma upregulated lncRNA) increases lipid biogenesis and lipid droplets in infected cells, which facilitates HCV release [114]. LncACOD is induced by multiple viruses, but not by type I interferon, and facilitates viral replication by binding to GOT2 (glutamic-oxaloacetic transaminase 2), increasing the catalytic activity of the enzyme and activating cellular metabolism [115]. As expected, most IFN-independent viral-induced lncRNAs function to facilitate viral replication or to modulate the transition between the latent and active phases of the viral life cycle. One exception to this is the HCV-induced lncRNA GAS5 (growth arrest-apecific 5), which binds and blocks the action of the nonstructural protein 3 (NS3) protease, and decreases HCV replication [110].

### 3.2. Virus-Induced lncRNAs Related to the Antiviral Response

Type I IFN-related lncRNAs are those whose function is relevant for the IFN response and/or whose expression is regulated after infection with several viruses or after cell treatment with PAMPs, DAMPs, or type I IFNs (Table 2). Indeed, several transcriptomic studies have highlighted the considerable deregulation of lncRNAs observed after treating cells with PAMPs, DAMPs, or type I IFNs [117,118,119]. Indeed, many lncRNAs are considered robust ISGs. Out of those functionally characterized, some have been shown to work in the initial stages of IFN induction. In contrast, others have been described to exert regulatory functions in the IFN response that leads to cytokine and ISG production. While the precise mechanism of action of the vast majority remains to be further explored, the role of a few of these lncRNAs is well documented.

#### 3.2.1. LncRNAs Are Involved in Viral Recognition and Modulate the Activity of Cellular Sensors

PAMP and DAMP recognition by cellular sensors is one of the first events in identifying a viral infection and triggering the innate immune response that aims to clear the virus [120]. The relevance of these initial stages of viral recognition has made cellular sensors a significant focus of immunomodulatory research for the treatment of hepatitis B virus (HBV), HCV, HSV, HIV, Zika virus (ZIKV), and many others [121]. As expected, lncRNAs have been described to work at early stages of infection, interacting with or modulating the expression and activity of canonical nucleic acid sensors such as cGAS, RIG-I, MDA5, or the non-canonical sensor PKR.

##### LncRNAs Can Regulate the Activity of Specific Nucleic Acid Sensors

Examples of such regulation are lncRNA ITPRIP-1 (inositol 1,4,5-triphosphate receptor interacting protein) and Lnczc3h7a, which work to enhance the activity of MDA5 and RIG-I respectively, and lnc-Lsm3b or LncATV, which block RIG-I signaling. HCV can induce ITPRIP-1 in a viral load- and time-dependent manner [122]. ITPRIP-1 was able to increase the expression of MDA5, MAVS, and the activation of IRF3, and affect HCV replication. RNA immunoprecipitation (RIP) analyses showed that ITPRIP-1 could bind MDA5 and promote its oligomerization and activation in a K63-linked polyubiquitination-independent manner. Further studies with a truncated MDA5 mutant showed that this partially explains the effect of ITPRIP-1 overexpression on IFN pathway activation. Surprisingly, this is not the only way ITPRIP-1 can regulate HCV replication through MDA5. It had been previously described that MDA5 could suppress viral replication by itself through binding to viral RNA [123]. In this case, it was shown that MDA5 could indeed bind the HCV genome, and ITPRIP-1 improved this interaction. Therefore, ITPRIP-1 can facilitate HCV clearance by an IFN-dependent and independent mechanism through binding to MDA5.

The positive action of Lnczc3h7a on RIG-I involves K63-linked polyubiquitination by the E3 ubiquitin ligase TRIM25 (tripartite motif-containing protein 25), which is required for RIG-I activity [124]. Lnczc3h7a forms a stable trimeric complex by binding to TRIM25 and RIG-I that allows TRIM25 activity and RIG-I signaling. In the case of LncATV, binding to RIG-I results in the repression of RIG-I antiviral signaling and, therefore, in a defective IFN response [125]. Indeed, LncATV is a cytosolic lncRNA whose expression responds to type I and type III interferon treatment (IFN-α2b and IFN-λ1) as well as viral infections with several viruses. Similarly, lnc-Lsmb3 impedes RIG-I signaling by competing with viral RNA for binding to RIG-I [126]. Binding of Lsmb3 to RIG-I monomers works as a proviral factor by avoiding activation of RIG-I, while at the same time, stabilizing inactive RIG-I. This works as a feedback stimulus to prevent the downstream signaling and late-stage production of type I IFNs.

The ability to titrate the concentration of effector molecules by competitive binding is a prominent mechanism of action of lncRNAs. This competition has been described for other ncRNAs as is the case of competing-endogenous RNAs (ceRNA) that work as miRNA sponges, but also as titrators of proteins. An excellent example of a competitive lncRNA is nc886 (pre-miR886 or vtRNA2-1) [127]. Upon infection, PKR sensing of viral dsRNAs induces PKR dimerization and kinase activity. Relevant PKR targets are eIF2a, which causes translation inhibition and apoptosis, and IκBa, which induces the NF-κB signaling pathway, leading to cell survival. This dual function of PKR plays relevant roles in cancer, and it is modulated by nc886 [128]. Similar to virus-associated RNAs from adenovirus, nc886 can bind to PKR with comparable affinity to dsRNA [129] and can block PKR kinase activity [130]. Thus, nc886 may play relevant roles in viral infection, cancer, and cell homeostasis, as it has been suggested that nc886 could control PKR activation in the absence of infection. Surprisingly, nc886 is required for PKR phosphorylation during T cell stimulation, contributing to the antiviral response at different levels [131].

##### LncRNAs Can Modulate the Levels of Nucleic-Acid Sensors

So far, we have described lncRNAs that preferably target PKR, RIG-I, or MDA5 through different mechanisms as a general response to viral infection. However, it is possible for a lncRNA to modulate this response in a cell-state-specific manner, as is the case of lncRHOXF1 (lncRNA Rhox homeobox family member 1). This lncRNA is abundantly expressed in trophectoderm and endoderm cells of human blastocyst-stage embryos [132]. Sendai virus (SeV) infection of human trophectoderm progenitor cells increased lncRHOXF1 transcription. In contrast, siRNA disruption during infection reduced the expression of viral sensor genes RIG-I and MDA5, possibly leading to the downregulation of downstream responders of IFN induction such as IFN-β, MxA, OAS1, and IFIT1 (Interferon-Induced Protein with Tetratricopeptide Repeats 1). This suggests that, while very specific in cell type and developmental stage, lncRHOXF1 can work as a general repressor of the antiviral response during early human development.

Instead, NEAT1 is a global regulator that directly impacts innate immune activation during viral sensing and downstream events. Thus, NEAT1 depletion increases the replication of several viruses, including HIV and HSV, and decreases lupus symptoms in mice [133,134,135]. Recently, it was shown that NEAT1 was induced by Hantaan virus (HTNV) through the RIG-I-IRF7 pathway in a time- and dose-dependent manner and promoted HTNV-induced IFN production by facilitating RIG-I and DDX60 expression [136]. DDX60 is a novel non-RLR helicase that works as a sentinel for the antiviral innate immune response in the cytoplasm, mostly in collaboration with RIG-I [137]. In addition to its role in viral sensing, NEAT1 was initially described to modulate cytokine production by favoring the sequestration of transcription factors in paraspeckles. Induction of NEAT1 by IAV, HSV-1, and by polyinosinic:polycytidylic acid (p(I:C))-triggered TLR3 pathway stimulation, results in increased formation of paraspeckles [138]. While the function of paraspeckles is not yet fully understood, it is reasonable to hypothesize that they can regulate gene expression by affecting the relocalization of transcription factors between nuclear sub-compartments. In fact, the increase in paraspeckle formation promoted the retention of the splicing factor proline/glutamine-rich (SFPQ), a NEAT1-binding protein. SFPQ works as a repressor of transcription of the IL-8 gene (*CXCL8*). Upon viral infection, NEAT1 induction relocates SFPQ from occupying *CXCL8* promoter to paraspeckles, leading to the transcriptional activation of *CXCL8*. Interestingly, this is the same mechanism proposed for NEAT1 control over RIG-I and DDX60. Finally, in addition to this transcriptional control, NEAT1 controls viral sensing more directly. NEAT1 binds to HEXIM1 (hexamethylene bis-acetamide-inducible protein 1) to assemble the HDP complex (HEXIM1-DNA-PK-paraspeckle), which also contains DNA-PK subunits (DNA-PKcs, Ku70, and Ku80) and several paraspeckle proteins [139]. HDP complex is a nuclear regulator of DNA sensing by cGAS. Binding of HDP to cGAS is remodeled by foreign DNA, leading to the release of paraspeckle proteins, recruitment of STING, activation of DNAPKcs, and IRF3 and IFN synthesis.

#### 3.2.2. LncRNAs Modulate Transcription and Transcription Factors of the IFN Synthesis Pathway

Nucleic acid sensors bind to cellular adaptors (TRIF, MyD88, and others) that activate several kinases, including TBK1 and IKK, in charge of phosphorylating and activating IRF3 and NF-κB, key transcription factors in the IFN synthesis pathway. LncRNAs regulate TRIF, MyD88, TBK1, IRF3, and NF-κB.

##### LncRNAs Can Regulate Cellular Adaptors and Modulate IRF3 Phosphorylation

LncRNA-155 was recently described to work in association with the protein effector tyrosine phosphatase 1B (PTP1B) [140]. LncRNA-155 is induced by several viruses such as IAV, SeV, muscovy duck reovirus (MDRV), and HSV via RIG-I and TLR3-dependent innate immune signaling, and disruption of lncRNA-155 expression in A549 cells impairs the antiviral response against IAV. LncRNA-155 is encoded within the *microRNA 155 host* gene (*MIR155HG*), whose deficiency in mice suppresses the production of immune cytokines and ISGs. This explains the increased susceptibility to IAV infection observed in these animals. Since MIR155HG also encodes the microRNA miR-155, experiments were developed that confirm that lncRNA-155 acts as a novel lncRNA involved in the regulation of innate antiviral immunity beyond the processing miR-155. Indeed, a miR-deleted lncRNA-155 also causes a significant decrease in viral loads and higher levels of IFN-β and MxA, IFIT1, ISG15, IFI27, OAS3, and IFITM3 after IAV infection. Mechanistically, miR-deleted lncRNA-155 inhibits PTP1B, a negative regulator of insulin and other signaling pathways. PTP1B inhibits activation by MyD88 and TRIF and affects MAPK, NF-κB, IRF3, and IFN responses [141]. Besides, PTP1B dephosphorylates IFNAR1 at Y466, favoring ligand-mediated receptor endocytosis, and decreasing IFN signaling [142]. Thereby, by inhibiting PTP1B, lncRNA-155 potentiates the IFN synthesis and signaling pathways.

MaIL1, induced by lipopolysaccharide (LPS)and TLR4 signaling and probably other PAMPs, is also a positive regulator of IFN synthesis. MaIL1 binds to optineurin (OPTN), a multifunctional ubiquitin reader, increasing OPTN stability. MaIL1 interaction helps the formation of ubiquitin-OPTN-TBK1 structures that facilitate IRF3 phosphorylation and activity [143]. IRF3 activity is also increased by lncLrrc55-AS, an IFN-induced lncRNA [144]. The role of lncLrrc55-AS in the IFN pathway has been well studied. lncLrrc55-AS binds the phosphatase methylesterase 1 (PME-1) and helps PME-1 binding to the protein phosphatase 2A (PP2A), which decreases IRF3 phosphorylation and signaling. Thus, lncLrrc55-AS increases the activity of PME-1, demethylation, and inactivation of PP2A and IRF3 activity.

##### LncRNAs Modulate NF-κB Activity Affecting the Production of Proinflammatory Cytokines

Similar to IRF3, several factors, including lncRNAs, are known to regulate NF-κB function. Several lncRNAs have been described to work in association with NF-κB with repercussions in inflammation-related diseases such as cancer, arthritis, cardiac disease, or diabetes [145]. In some cases, these lncRNAs have been shown to regulate NF-κB in response to bacterial PAMPs or other stresses that lead to inflammation. Although still unclear, it is reasonable to predict that these lncRNAs could also play antiviral functions. For instance, THRIL (TNF-α and hnRNPL-related immunoregulatory lincRNA) is one of the many lncRNAs induced after TLR1/2 activation [146]. THRIL forms a complex with heterogeneous nuclear ribonucleoprotein L (hnRNPL) that binds to the *TNF-α* (*tumor necrosis factor alpha*) promoter to induce its transcription. Using a similar mechanism, lincRNA-EPS serves as a scaffold for hnRNPL to maintain a repressive chromatin state at the transcription start sites of many immune response genes, until it becomes downregulated after TLR4 stimulation with LPS. Indeed, knockout of lincRNA-EPS in mice led to enhanced inflammation and increased lethality following endotoxin challenge in vivo [147]. Other lncRNAs target NF-κB subunits instead of chromatin. For example, p50-associated cyclooxygenase-2 extragenic RNA (PACER) functions as a positive regulator of NF-κB by obstructing p50 repressor homodimers in inflammation-driven tumors [148]. NF-κB interacting lncRNA (NKILA) directly blocks IκB phosphorylation to form a stable ternary complex NF-κB/IκB/NKILA that represses cancer-associated inflammation in breast cancer [149]. Similarly, Lethe is a TNF-α-induced pseudogene that binds to NF-κB p65/RelA subunit and blocks binding to DNA, leading to decreased inflammation [150]. Finally, lnc-EPAV (endogenous retrovirus-derived lncRNA) is induced after infection with RNA viruses or treatment with RNA mimics [151]. Lnc-EPAV functions by competitively binding to RelA and displacing its transcriptional repressor SFPQ. This leads to increased levels of RelA and its proinflammatory targets.

##### LncRNAs Control IFN Transcription

While modulation of the activity of IRF3 and NF-κB by lncRNAs impacts strongly on IFN-β transcription, few studies describe a direct lncRNA regulation of the transcription of type I IFNs. Instead, INF-γ transcription is strongly dependent on the expression of lncRNA NEST/TMEVPG1 (nettoie salmonella pas theiler’s)/(theiler’s murine encephalomyelitis virus persistence candidate), located in the vicinity of the *IFNG* gene [152,153,154]. An interesting regulator of IFN-β transcription is Lnc-MxA, which functions by forming RNA-DNA triplexes with the promoter of IFN-β. Lnc-MxA had been previously described to be upregulated during IAV infection [155]. Overexpression of lnc-MxA helps viral replication, whereas knockdown of lnc-MxA inhibits replication of IAV, SeV, and VSV. Mechanistically, it was shown that the RNA-DNA triplex formed at the IFN-β promoter hindered the binding of IRF3 and NF-κB, effectively inhibiting its transcription and downstream activation of type I IFNs.

#### 3.2.3. LncRNAs Are Involved in IFN Signaling

Binding of IFN to the IFNAR triggers phosphorylation of STAT1 and STAT2 and induces MAPK and PI3K signaling. Few lncRNAs have been described to date that affect the IFN signaling pathway directly. LncRNA-155 regulates IFNAR endocytosis (see above). Further down in the pathway, lncRNA Morrbid (myeloid RNA repressor of BCL2L11 induced death) modulates the strength of the PI3K pathway and regulates the expression of the proapoptotic factor Bcl-2-like protein 11 (Bcl2l11) [156]. Morrbid can be induced by infection, IFN stimulation, or T-cell receptor activation of CD8+ T-cells and contributes to cell expansion, survival, and effector functions.

Instead, regulation of STAT transcription factors seems to be a significant target for lncRNA-mediated control. Such regulation has been described at different levels. Lnc13 loci can harbor an SNP (single nucleotide polymorphism) associated with type I diabetes [157]. Samples with the risk genotype show higher levels of STAT1 than heterozygous samples. Mechanistically, p(I:C) induction increases the translocation of lnc13 from the nucleus to the cytoplasm, where it favors STAT1 mRNA binding to poly(rC) binding protein 2 (PCBP2) and results in increased STAT1 mRNA stability. In the same line, lncRNA Sros1 blocks the binding between STAT1 mRNA and cell cycle-associated protein 1 (CAPRIN1), which contributes to STAT1 mRNA stabilization and signaling [158]. Although Sros1 activity has been described after stimulation with IFN-γ, it could also regulate STAT1 expression in the context of type I IFN signaling. Similarly, the activity of lncRNA625 has been shown in untreated and IFN-γ-treated cells [159]. LncRNA625 interacts with the DNA binding domain of STAT1 and contributes to the binding between STAT1 and the T-cell protein tyrosine phosphatase TC45, which dephosphorylates STAT1 and impedes STAT1-mediated transcription of ISGs. RP11-2B6.2, a lncRNA upregulated in the kidney of SLE patients with nephritis, also affects STAT1 phosphorylation. RP11-2B6.2 decreases chromatin accessibility and transcription of SOCS1, a negative regulator of STAT1, leading to increased IFN response [160]. Less clear is the mechanism by which linc00513 favors phosphorylation of STAT1 and STAT2 [161]. Linc00513 is interesting because it is overexpressed in SLE patients and contains functional susceptibility loci in the promoter region. Two SNPs enhanced lnc00513 promoter activity, increased lncRNA levels, and IFN response.

#### 3.2.4. LncRNAs Can Modulate the Transcription of Specific ISGs

Activation of the IFN synthesis and signaling pathways leads to a strong induction in the transcription of ISGs and inflammatory genes. Many lncRNAs are known to directly impact ISG expression, even in the absence of upstream regulation from viral sensors or IFN induction intermediates. This control may be more general or may be exerted over specific genes.

Transcriptomic analysis comparing IFN-α-treated and control cells show a massive deregulation of the coding and non-coding genome [102,118]. Interestingly, many lncRNAs upregulated after IFN-α treatment, are located near (convergent, divergent, or in tandem) to coding ISGs or inflammatory genes. It has been hypothesized that these lncRNAs could act in *cis* and regulate the expression of their coding neighbors. Indeed, many coding and non-coding pairs are significantly co-expressed. However, co-expression could be the result of co-regulation, which is most probably occurring for divergent coding/non-coding pairs sharing a bidirectional promoter. Therefore, as of today, it cannot be concluded that a general function of IFN-induced lncRNAs is to regulate the expression of their neighboring coding genes.

Nevertheless, several IFN-induced lncRNAs have been shown to regulate the levels of their neighboring ISG. Lnc-ISG20 shares most of its sequence with ISG20, and both show a similar response to IFN [155]. Lnc-ISG20 is a positive regulator of ISG20 and can inhibit IAV replication in an ISG20-dependent manner. Lnc-ISG20 acts as a ceRNA by binding to miR-326 and releasing ISG20 from miR-326-mediated downregulation. Indeed, several lncRNAs titrate the levels of other ncRNAs to exert their regulatory functions, usually in lncRNA-miRNA-mRNA networks, frequently working as ceRNAs [162]. Instead, IFN-induced lncRNA BISPR (bone marrow stromal cell antigen 2 (BST2) interferon-stimulated positive regulator), transcribed from a bidirectional promoter shared by coding gene BST2, is a positive regulator of BST2 expression at the transcriptional level [163,164]. As BST2/tetherin prevents diffusion of viral particles after budding from infected cells, the downregulation of BISPR leads to a strong decrease in hepatitis E virus egress [165]. IFN-induced lncRNA IRF1-AS activates IRF1 transcription by binding to ILF3 (interleukin enhancer binding factor 3) and DExH-box helicase 9 (DHX9) [166]. In turn, IRF1 binds to the IRF1-AS promoter directly and activates IRF1-AS transcription, leading to a positive regulatory loop. Neighboring lncRNAs can also function as negative regulators. IFN-induced lncRNA-IFI6 (interferon alpha inducible protein 6) is antisense and within the first intron of IFI6 and inhibits IFI6 transcription by promoter histone modification, leading to increased HCV replication [167].

While the deposition of H3K4me3 marks at promoters is essential for priming immune genes for expression, it is unclear how this is achieved at the single-gene level. Neighboring lncRNAs could play this role. LncRNA UMLILO (upstream master lncRNA of the inflammatory chemokine locus) functions in *cis* to attract WDR5-MLL to neighboring chemokine promoters located within the same topologically associated domain (TAD) [168]. Then, MLL1 facilitates H3K4me3 deposition at these promoters and epigenetic priming. Interestingly, UMLILO is just the prototype of a collection of immune gene-priming lncRNAs that functions in combination to establish trained immunity.

#### 3.2.5. LncRNAs Can Be General Regulators of the Transcription of Several ISGs and Inflammatory Genes

##### LncRNAs Can Negatively Regulate the Transcription of ISGs and Help Viral Replication

Negative regulator of interferon response (NRIR) was the first example of lncRNA that regulates the expression of several ISGs in the nucleus by affecting ISG transcription [118]. Located near the ISG *CMPK2* (cytidine/uridine monophosphate kinase2), lncRNA-CPMK2 or NRIR was found upregulated upon treatment with IFN-α and IFN-γ, and to be a bona fide ISG [118]. NRIR decreases the expression of a subset of ISGs located near NRIR loci, CMPK2 or viperin, or far away in the genome, such as IFIT3, IFIT1, CXCL10 (C-X-C motif chemokine 10), or ISG15. Interestingly, NRIR depletion increases ISG expression in both control and IFN-stimulated cells, indicating that it is a genuine negative regulator of ISG expression. The mechanism for this regulation was proposed to work at the transcriptional or epigenetic level, mainly because the stability of the ISG messengers remained unchanged in the absence of NRIR.

Similarly, TSPOAP1-AS1 (translocator protein-associated protein 1-antisense 1), induced by IAV and p(I:C), accumulates in the nucleus after infection, inhibiting IFNB1 transcription and ISG expression, effectively helping viral replication [169]. Some authors have described lncRNAs that affect histone modification of ISG promoters by unknown mechanisms. NRAV (negative regulator of antiviral response) is a classic example of a lncRNA that responds to infection with several viruses, and that has a significant impact on the fate of the infected cell by altering its ability to mount an active ISG-mediated antiviral state [170]. NRAV negatively regulates the expression of several critical ISGs such as IFIT2, IFIT3, IFITM3, OASL, and MxA, impacting IAV replication in human cells and IAV virulence in transgenic mice. Mechanistically, NRAV reduced the initial transcription rates of MxA and IFITM3 by altering their histone modifications (active mark H3K4me3 and repressive mark H3K27me3). ZONAB (Zo-1-associated nucleic acid binding proteins), a well-known transcriptional regulator of cyclin D1 and PCNA, was identified as an NRAV-associated protein involved in MxA transcription regulation. However, it is still unclear whether ZONAB can function as a general transcription factor for ISG expression.

##### LncRNAs Can Positively Regulate the Transcription of ISGs and Prevent Viral Replication

Since NRIR was described, many lncRNAs have been shown to alter the levels of several ISGs, most likely at the transcriptional level. In most cases, the molecular mechanisms are unknown. This is the case of metastasis-associated lung adenocarcinoma transcript 1 (MALAT1), which contributes to the expression of OAS2, OAS3, and OASL and controls IFN-α response in SLE [171]. Several hnRNPs have been shown to bind different lncRNAs that regulate the transcription of inflammatory genes and ISGs. It is currently unknown whether these hnRNPs function to stabilize or guide the lncRNA to the proper cellular context or if they are adaptors that help interaction between the lncRNA and transcription regulators. Binding to hnRNPU was described to mediate the chromatin state changes behind IVRPIE (inhibiting IAV replication by promoting IFN and ISG expression) regulation of the antiviral response during IAV infection [172]. IVRPIE is upregulated in patients with acute IAV and limits IAV replication by increasing mRNA and protein levels of IFN-β1, and some critical ISGs, including IRF1, IFIT1, IFIT3, MxA, ISG15, and IFI44L. ChIP (chromatin immunoprecipitation) experiments showed that IVRPIE expression correlated with increased activating H3K4me3 and decreased repressing H3K27me3 marks from the TSS of some of these genes. hnRNPU is required to activate these genes, except ISG15, but a clear connection between hnRNPU and epigenetic modifiers has not yet been established.

An exception to the positive or negative regulator dichotomy is lncCOX2 (long non-coding cyclooxygenase 2b), which is induced by Myd88 and NF-κB and mediates both the activation and repression of immune genes [173]. Transcriptional downregulation of inflammatory genes and ISGs requires binding between lncCOX2 and hnRNPA/B and hnRNPA2/B1. While the exact mechanism of action is unknown, knocking down these hnRNPs or lncCOX2 increases the recruitment of RNA pol II to the promoter of target genes. Also, after LPS stimulation, lncCOX2 can bind the SWI/SNF (SWItch/sucrose non-fermentable) chromatin remodeling complex and helps for NF-κB transcription of late-primary inflammatory response genes in macrophages responding to microbial challenge [174]. Additional studies have described that lncCOX2, located in both the nucleus and the cytoplasm, can also bind p65/RelA and promote its nuclear translocation and transcriptional activation of the inflammasome sensor NLRP3 and adaptor protein ASC (apoptosis-associated speck-like protein) [175].

##### Some lncRNAs Can Regulate ISGs by Targeting Specific Transcriptional Regulators

Several lncRNAs have been described to regulate antiviral genes by modulating the function of specific transcription regulators. In the case of lncITM2C-1 (lncRNA integral membrane protein 2C), the mediator is the neighboring gene *G protein-coupled receptor 55* (*GPR55*), a cannabinoid receptor with anti-inflammatory properties [176]. lncITM2C-1 is induced by HCV infection and helps viral replication by contributing to GPR55 expression [177]. The mediator of Loc107051710 is IRF8, a transcription factor that participates in the expression of immune and antiviral genes, and that belongs to the same family of IFN-induced transcription factors as IRF1, IRF3, or IRF7 [178]. After infectious bursal disease virus (IBDV) infection, Loc107051710 expression is induced. It translocates from the nucleus to the cytoplasm and activates IRF8-mediated expression of type I IFN, STATs, and ISGs, leading to decreased viral levels [179]. Similarly, lncRNA#32 or lncRNA upregulator of antiviral response interferon signaling (LUARIS), can suppress HBV and HCV infection by binding to activating transcription factor 2 (ATF2) and regulating ISG expression in human hepatocytes [180]. LUARIS is induced by p(I:C) and contributes to the expression of known ISGs and chemokines, including IRF7, OASL, RSAD2 (radical S-adenosyl methionine domain containing 2), CCL5 (chemoquine (C-C motif) ligand 5), CXCL10, and CXCL11. Interestingly, this effect is independent of IFN-β treatment, suggesting that LUARIS regulates the level of ISG and chemokine expression by itself under unstimulated and IFN-stimulated conditions. This effect is helped by the interaction of LUARIS with ATF2 and hnRNPU, which increased its half-life. Indeed, the knockdown of ATF2 reduced the levels of LUARIS-regulated ISGs.

More canonical transcription regulators mediate the activity of AK006025, EGOT (eosinophil granule ontogeny transcript), and lnc13. Mouse astrocytes treated with the HIV Nef protein upregulate the levels of AK006025 and other lncRNAs predicted to function in inflammation [181]. AK006025 associates with CREB-binding Protein/p300 (CBP/p300), contributes to the increased acetylation in the promoter of CXCL9, 10, and 11 genes, and causes the increased expression of these genes. EGOT is a lncRNA highly induced by infection with RNA viruses or RNA mimics and by the activation of different stress-response pathways, including NF-κB and PI3K/AKT [102,182]. Under conditions of HCV infection, EGOT further inhibits the low levels of several ISGs contributing to HCV replication [102]. However, in non-infected cells, EGOT potentiates the activity of NF-κB by controlling the levels of the transcriptional coactivator T-Box protein 1 (TBXL1) [182]. Under this condition, EGOT inhibition leads to decreased TBLX1 levels and decreased transcription of immune factors and ISGs. It is unclear how EGOT, a *cis*-acting lncRNA, can modulate the expression of TBLX1. A clearer picture can be drawn for lnc13, previously described as a STAT1 mRNA regulator. lnc13 is a chromatin-bound lncRNA whose expression decreased after LPS stimulation in an NF-κB-dependent manner [183]. Lnc13 decreases transcription of specific inflammatory genes such as MyD88, STAT1, STAT3, or IL1RA. Mechanistically, lnc13 binds to hnRNPD, an inflammation-related factor that can regulate gene expression by binding to the nucleosome remodeling complex, NuRD (nucleosome remodeling deacetylase). In agreement with this possibility, lnc13 binds to HDAC1, a component of the NuRD complex. Downregulation of lnc13 or hnRNPD shows similar effects on the expression of inflammatory genes. Further, lnc13, HDAC1, and hnRNPD bind to the same chromatin regions. Interestingly, the downregulation of lnc13 or hnRNPD decreases the binding of HDAC1 to these regions. These results agree with lnc13 action as a linker between chromatin and hnRNPD–HDAC1 complex to decrease the expression of specific inflammatory genes. Interestingly, lnc13 levels are decreased in intestine biopsies from patients with celiac disease, where lnc13 downregulation could contribute to inflammation. In favor of this possibility, disease-associated variants of lnc13 bind hnRNPD less efficiently than wild-type versions. Therefore, decreased binding to hnRNPD of lnc13 mutants may lead to increased levels of inflammatory genes and contribute to celiac disease.

**Table 2 ijms-21-06447-t002:** LncRNAs deregulated by viral infections and involved in the innate antiviral immune response.

LncRNA	Pathway	Stimuli	Study Design	Role	Mechanism of Action	References
ITPRIP-1	IFN-α	HCV	HLCZ01 (HCC) +/− IFN-α	AV	Binds, stabilizes, and activates MDA5, and improves direct binding of MDA5 to HCV genome	[122]
Lnczc3h7a	type I IFN	VSV, SeV	RAW264.7 (mouse Mφ) +/− VSV	AV	Forms a trimeric complex with TRIM25 and RIG-I that allows RIG-I signaling	[124]
LncATV	IFN-α,β,λ	HCV, ZIKV, NDV, SeV	HuH7 (HCC) +/− IFN-α2b and IFN-λ1	PV	Binds RIG-I and prevents RIG-I signaling	[125]
Lnc-lsm3b	type I IFN	VSV	RAW264.7 (mouse Mφ) +/− VSV	PV	Competes with viral RNA for binding to RIG-I, stabilizing inactive RIG-I and preventing RIG-I activation	[126]
nc886	eIF2-α, NF-κB	Adenovirus	CD4+ T cells +/− TCR activation	PV	Binds PKR and blocks PKR-mediated activation of eIF2-α and NF-κB pathways	[128,131]
lncRHOXF1	IFN-β	SeV	Human trophoblasts +/− SeV	PV	Decreases the expression of RIG-I and MDA	[132]
NEAT1	Type I IFN	HTNV, IAV, HSV	HUVECs +/− HTNV. Hela and A549 +/− p(I:C), IFN-α, IFN-β	AV	Relocates the transcriptional repressor SFPQ to paraspeckles, favoring the expression of RIG-I, DDX60, IFN-β, and IL-8	[136,138]
LncRNA-155	IFN-β	IAV, SeV, MDRV, HSV	C57BL/6J mice lungs or A549 +/− IAV	AV	Inhibits PTP1B expression, resulting in higher production of IFN-β and critical ISGs including Mx1, IFIT1, ISG15, OAS3	[140]
lncLrrc55-AS	type I IFN	VSV, SeV HSV, LPS, p(I:C), IFN-α, IFN-β	WT and IFNα/β KO Mφ +/− VSV	AV	Supports PME-1-mediated inactivation of PP2A, enhancing IRF3 phosphorylation, activation, and IFN production	[144]
lnc-EPAV	NF-κB	p(I·C), SeV and VSV	Mouse BMDMs from C57BL/6 +/− p(I:C)	AV	Promotes expression of RELA by competitively binding to and displacing its transcriptional repressor SFPQ	[151]
NeST/IFNG-AS1/ TMEVPG1	IFN-γ	TMEV	B10.S and SJL/J mice +/− TMEV	AV	Binds WDR5 to deposit activating epigenetic marks on the IFN-γ promoter of infection-resistant immune cells	[184]
Lnc-MxA	IFN-β	IAV, SeV, p(I:C) and IFN-β	A549 and HEK293T +/− IAV	PV	Forms an RNA-DNA triplex with the IFN-β promoter that prevents the binding of IRF3 and NF-κB	[185]
Morrbid	Type I IFN	LCMV, TCR + IFN-α or IFN-β	CD8+ T cells +/− LCMV	PV	Controls CD8 T cell expansion, survival, and effector function by regulating the expression of Bcl2l11	[156,186]
Lnc13	STAT1	p(I:C) and CVB5	Expression screen in EndoC-βH1 cells	AV	Interacts with PCBP2 to regulate the stability of the STAT1 mRNA, and the upregulation of CXCL10 and CCL5	[157]
Lnc-ISG20	type I IFN	IAV	A549 and HEK293T +/− IAV	AV	Binds miR-326 acting as a ceRNA and de-repressing the transcription of ISG20	[155]
lncRNA-BST2/BISPR	IFN-α IFN-λ	IAV, VSV, HCV, IFN-α, and IFN-λ	HuH7 +/− IFN-α2	AV	Activates the transcription of BST2 which prevents diffusion of viral particles after budding from infected cells	[163,164]
LncRNA-IFI6	IFN-α	HCV, and HCV-JFH1	Huh7.5.1 and PHH +/− IFN-α	PV	Negatively regulates IFI6 promoter by histone modification through a spatial structural domain	[167]
lncRNA-CMPK2/NRIR	IFN-α IFN-λ	HCV	PHH +/− IFN-α	PV	Negatively regulates the expression of several ISGs	[118]
TSPOAP1-AS1	IFN-β, NF-κB	IAV and p(I:C)	A549 +/− AIV	PV	Negatively modulates IAV-induced Ifnb1 transcription, ISRE activation, and downstream ISG expression	[169]
NRAV	type I IFN	IAV, SeV, MVRD, HSV	A549 +/− AIV	PV	Regulates the transcription of multiple ISGs, including IFITM3 and MxA, by affecting their histone modifications	[170]
IVRPIE	IFN-β	IAV, SeV, VSV and p(I:C)	PBLs +/− IAV infection	AV	Regulates the transcription of IFN-β1 and ISGs, including IRF1, IFIT3, MxA, and ISG15, by epigenetic modification	[172]
lncCOX2	NF-κB	LPS, p(I:C) and TMEV	BV2 (mouse microglia) +/− LPS	AV	Coactivator of NF-κB for the transcription of immune response genes through epigenetic remodeling by SWI/SNF	[174]
lncITM2C-1	Type I IFN	HCV, p(I:C)	Huh-7.5 +/− miR-122 and/or HCV	PV	Stimulates the expression of neighboring gene GPR55, which in turn downregulates ISGs such as ISG15, Mx1, and IFITM1	[177]
Loc107051710	Type I IFN	IBDV	DF-1 (chicken fibroblasts) +/− IBDV	AV	Promotes the production of IFN-α and IFN-β by regulating IRF8, thereby promoting ISGs antiviral activity	[179]
LUARIS/lncRNA 32	IFN-β	EMCV, HBV and HCV	HuS IRF3 KO cells +/− IFN-β	AV	Binds ATF2 to regulate the expression of chemokines such as IP-10 and CCL5	[180]
AK006025	NF-κB	HIV	Mouse astrocytes +/− HIV’s Nef protein	AV	Associates with NF-κB p65 and CBP/P300 to epigenetically regulate Nef-induced Cxcl9/10/11 cluster gene expression	[181]

IFN: interferon; HCC: hepatocellular carcinoma; ZIKV: zika virus; NDV: Newcastle disease virus; eIF2α: Eukaryotic translation initiation factor 2 alpha; NF-κB: nuclear factor kappa b; TCR: T-cell receptor (αCD3/αCD28); HTNV: hantaan virus; HUVECs: human umbilical vein endothelial cells; p(I:C): polyinosinic: polycytidylic acid; MDRV: muscovy duck reovirus, LPS: lipopolysaccharide; KO: knock-out; BMDMs: Bone-marrow-derived macrophages; TMEV: Theiler’s murine encephalomyelitis; LCMV: lymphocytic choriomeningitis mammarenavirus; CVB5: Coxsackie Virus B5; JFH1: Japanese fulminant hepatitis 1; PBLs: Peripheral blood leucocytes; miR-122: microRNA 122, an essential miRNA for HCV replication; IBDV: infectious bursal disease virus; DF-1: chicken embryo fibroblasts;; EMCV: encephalomyocarditis virus; HuS: immortalized human hepatocytes; LC: Lung carcinoma;;; ESCC: esophageal squamous cell carcinoma; PHH: primary human hepatocytes;;; PV: proviral; AV: antiviral.

## 4. Concluding Remarks and Future Perspectives

IFN-related lncRNAs play fundamental roles as inducers or repressors in almost every step of the IFN response. Certainly, the number of lncRNAs described to date to work as regulators of the antiviral system seems quite remarkable. Although lncRNA genes are numerous and more are expected to be described in the near future, deciphering the molecular mechanisms that drive lncRNA function involves an arduous and often cumbersome work. Indeed, there are many virus-regulated lncRNAs whose role in viral replication or viral clearance has been robustly validated but are still being mechanistically interrogated. It is unclear nowadays whether lncRNAs related to IFN are significantly more compared to other fields. However, this would not be surprising as lncRNAs are fast evolving molecules that could be acting especially in processes under high evolutionary pressure, as is the firing of the antiviral response and its subsequent self-limitation.

LncRNA regulation of the immune response is not limited to their interaction with canonical factors of the IFN pathway. In fact, the role played by metabolism in the antiviral response has been elucidated, in part, thanks to the study of virus-related lncRNAs. An active metabolism is required to produce the energy and resources required for the synthesis of viral components. As immune activation is also dependent on such resources, the metabolic regulation of infected cells needs to be tightly controlled, a new field of function of lncRNAs that is expected to grow in the coming years. Other lncRNAs expected to work in immunity include those that play a role in RNA editing. RNA modifications are being increasingly shown to regulate coding and lncRNAs and to play a significant role in immune activation and regulation. For example, m6A and pseudouridine modifications serve to mark endogenous RNAs and decrease their recognition by TLR sensors [187].

Although the mechanism of action of lncRNAs can be elusive, furthering research in their roles as central host factors in immunity, especially in the antiviral response, is an urgent matter given the strong impact that these factors may have on basic antiviral research and therapy in a longer term. Indeed, today’s research in IFN-related lncRNAs is the ground for the development of future therapies for viral infections, autoimmune diseases, and cancer. The coronavirus disease 2019 (COVID-19) pandemic that started at the end of 2019 and is expected to span through 2020 and further has shown the importance of advancing virus research. Less than two weeks after the first report of a cluster of pneumonia cases in Wuhan, China, the genetic sequence of SARS-CoV-2 was shared with the world, quickly followed by diagnostic protocols, which have helped to prompt clinical and public health interventions. Nonetheless, this pandemic has also exposed our limited ability to respond to viral outbreaks in terms of treatment, prediction of disease outcome, and response to therapy. This field could benefit significantly from lncRNA research, especially in the context of the IFN response. Multiple studies in humans and animal models have shown that corticosteroid immunosuppression hinders the induction of antiviral type I IFN in response to a wide range of respiratory viruses [188,189]. Targeted therapies against JAK-STAT signaling have shown similar results [190]. IFNs are part of treatment regimens to treat chronic viral infections. IFN-α has been used in the treatment of HBV, HCV, HIV, HSV, and IAV infection [191]. A combination of antiretroviral therapy (ART) with type I IFN is effective in reducing HIV and partially resurrects the immune system of the infected. However, in the absence of ART, IFN-α cannot limit viral replication by itself [192]. Unfortunately, adverse effects of IFN therapies impact most organ systems, and symptoms range from mild to severe, including anemia, skin conditions, autoimmune hepatitis, idiopathic thrombocytopenia, or pancreatitis [72]. On the other hand, systemic inflammation is known to drive the severity, progression, and overall outcome of many virus-related diseases, including COVID-19 [193,194]. Direct suppression of IFN and other pathway mediators such as IL-6 or TNF-α, in patients with cytokine storm and overwhelming viral illnesses, could become counterproductive when weighed up against the potentially disadvantageous effects of inhibiting antiviral immunity, delaying virus clearance and extending the infection.

Similarly, many autoimmune diseases, such as SLE, have been linked to an abnormal function of the IFN response. Therefore, the aberrant activity of lncRNAs that potentiate IFN pathways could be a risk for the development of interferonopathies. Indeed, a break in self-discrimination and tolerance to abnormally accumulated or chemically modified nucleic acids is one of the main mechanisms proposed as the molecular basis of type I interferonopathies. This heterogenous group of diseases is characterized by systemic autoinflammation, usually as a result of constitutively high levels of IFN-related activity in the absence of infection. Interferonopathies are induced by monogenic mutations, such as in the case of Aicardi-Goutières syndrome, or may be polygenic diseases, such as systemic lupus erythematosus (SLE). While there are no studies linking monogenic interferonopathies with lncRNAs yet, the impact on IFN-related lncRNA regulators on SLE is now well documented [160,171,183]. This is especially relevant since functional targets of IFN-related lncRNAs such as JAK/STAT, cGAS, STING, and RIG-I have also been proposed as targets for the immunomodulatory treatment of type I interferonopathies [195,196].

In addition, the IFN response is highly relevant for cancer development and progression. Chronic inflammation caused by different risk factors, including viral infections with HBV and HCV, lays the ground for tumorigenic events. Sensing of PAMPs and DAMPs initiates stress-induced pathways such as NF-κB, PI3K/AKT, or MAPK that may promote cell survival and proliferation, contributing to tumor growth. Further, IFN-mediated immunity has been related to the efficacy of checkpoint-blockade therapy [72]. For example, it has been recently described that IFN induces the *IFN-stimulated non-coding RNA 1* (*INCR1*) gene, located in the PD-L1 locus. LncRNA INCR1 binds hnRNPH1, preventing its negative effect on the expression of the neighboring genes PD-L1 and JAK2. This results in increased IFN-γ signaling and affects T cell-mediated clearance of tumor cells [197].

In these settings, lncRNAs surface as promising candidates for therapeutic interventions for several reasons: (1) they can target viral proteins directly and selectively to prevent viral replication, (2) they can fine-tune the immune response in specific stages of antiviral immunity activation, from nucleic acid-sensing or downstream targets of IFN induction to a broad or specific modulation of ISG production, (3) they can work or be induced independently of JAK-STAT signaling or IFN signaling altogether, meaning they could be used alternatively or in combination with JAK-STAT or IFN modulators, and (4) since they are very tissue- and cell-state-specific, their use in immune modulation could potentially restrict unwanted secondary effects of systemic immune suppression or activation. However, lncRNA-based therapies are still in their initial stages and will not be arriving in the clinic in the short term. Nonetheless, lncRNAs could also be used as biomarkers of disease severity, progression, and response to treatment. Many lncRNAs behave as acute-phase reactants that can be measured in body fluids and could potentially be used to inform decision-making on patient management and surveillance of disease. Progress in this field, however, is contingent on a substantial investment of resources into studies with larger cohorts of patients, best-in-class follow-up, and prospective analyses that can allow safe and efficient interrogation of lncRNAs potential as therapeutic targets and biomarkers for immune-related diseases.

## Figures and Tables

**Figure 1 ijms-21-06447-f001:**
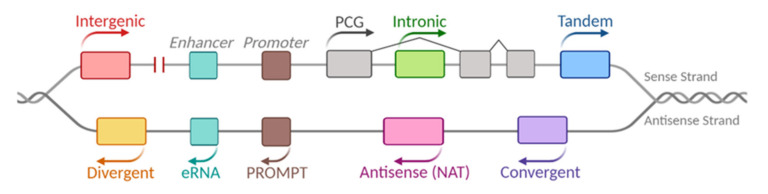
Long non-coding RNA (LncRNA) classification according to their genomic context relative to protein-coding genes (PCGs) and their regulatory DNA elements. See text for details.

**Figure 2 ijms-21-06447-f002:**
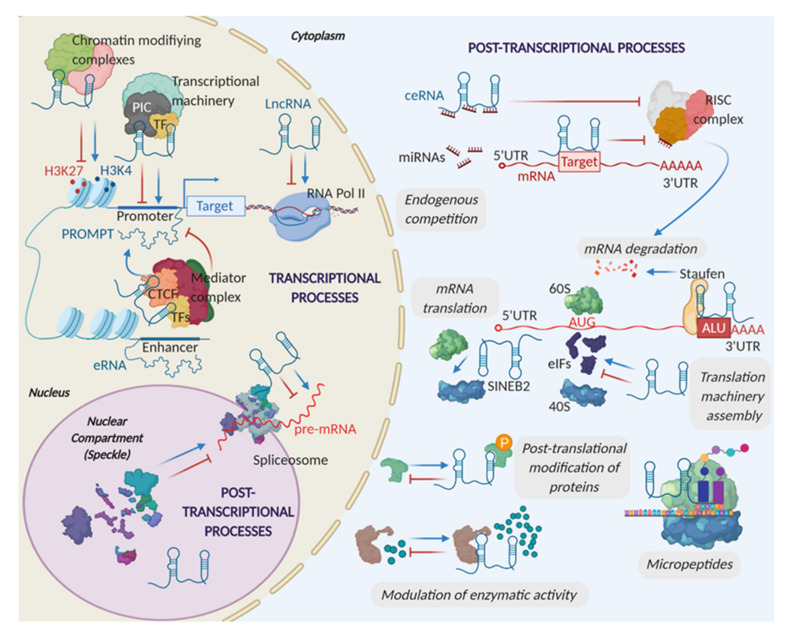
LncRNAs affect protein activity directly or by regulating transcriptional and post-transcriptional processes in the nucleus (on the left) or in the cytoplasm (on the right). These effects result from the ability of lncRNAs to form primary, secondary, and tertiary structures able to bind DNA, RNA, or proteins. See text for details. Blue arrows depict activation while the red arrows, inhibition.

**Figure 3 ijms-21-06447-f003:**
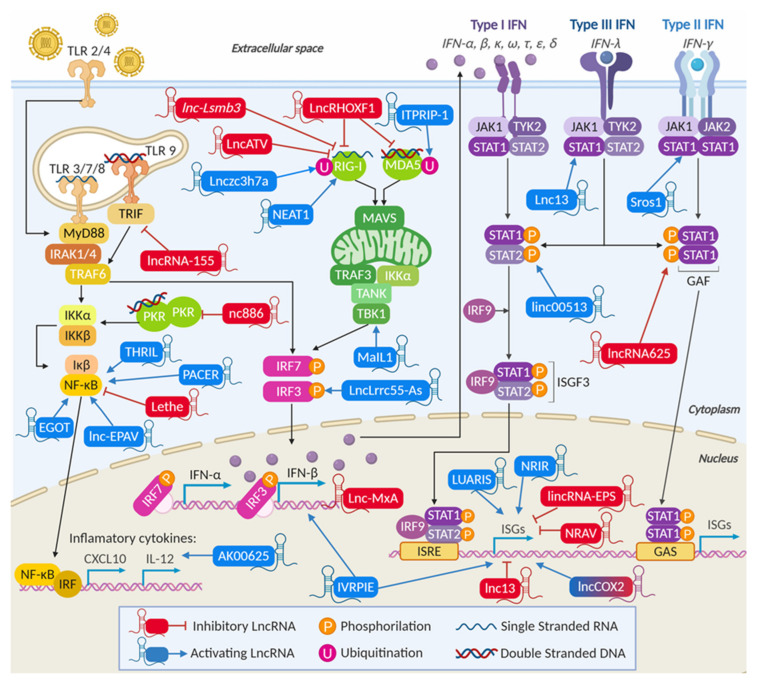
LncRNAs affect many steps of the interferon (IFN) synthesis (left) and signaling pathways (right). See text for details. LncRNA-linked blue arrows depict activation while red arrows indicate inhibition. Name frames are in blue for activating lncRNAs, red for inhibitory lncRNAs, and in both colors for lncRNAs with dual functions.

**Table 1 ijms-21-06447-t001:** LncRNAs deregulated by viral infections unrelated to the antiviral response.

LncRNA	Stimuli	Study Design	Role	Mechanism of Action	References
NRON	HIV	CD4 + T cells screen	AV	Links Tat to the ubiquitin/proteasome components CUL4B and PSMD11, thus facilitating Tat degradation	[109]
PAAN	IAV	HEK293T +/− IAV	PV	Promotes the assembly of viral RNA polymerase, warranting efficient viral RNA synthesis	[110]
IPAN	IAV	HEK293T +/− IAV	PV	Stabilizes viral RNA polymerase PB1, enabling efficient viral RNA synthesis	[111]
HEAL	HIV	MDMs +/− Mφ-tropic HIV-1	PV	Forms a complex with FUS, which facilitates HIV replication by recruiting p300 to the HIV promoter	[112]
MAMDC2-AS1	HSV-1	HDFn +/− HSV-1	PV	Interacts with Hsp90α to facilitate the nuclear transport of VP16, the core factor initiating the expression of HSV-1 IE genes.	[113,116]
HULC	HCV	Huh7.5 +/− HCV	PV	Manipulates the lipid pool to favor loading of HCV-core protein onto lipid droplets and subsequent virus-particle release	[114]
GAS5	HCV	Huh7 +/− HCV	AV	Inhibits viral replication by decoying HCV NS3 protein	[110]
ACOD	SeV, VSV, HSV-1, VACV	WT and *Ifnar*^−/−^ Mφ +/− VSV	PV	Facilitates viral replication by binding to GOT2, increasing the catalytic activity of the enzyme and activating cellular metabolism	[115]

HIV: human immunodeficiency virus; IAV: influenza A virus; MDMs: monocyte-derived macrophages; Mφ: macrophages; HSV-1: herpes simplex virus 1; HDFn: primary human fibroblasts; VP16: viral tegument protein; IE: immediate-early; HCV: hepatitis C virus; NS3: non-structural protein 3; SeV: sendai virus; VSV: vesicular stomatitis virus; VACV: vaccinia virus; WT: wildtype; PV: proviral; AV: antiviral.

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
