# Peer review of "LncRNAs in the Type I Interferon Antiviral Response"

_ijms, 2020, doi:10.3390/ijms21176447_

Round 1
Reviewer 1 Report
In this review, Suarez et al. perform and extensive revision of the literature and compile the main works done in the context of lncRNAs and type I interferon antiviral response. The review is well organized and the tables and the figures are very informative. My only concern is that some parts are long and harbor a lot of information making it a little too dense. If possible I would try to subdivide the sections 3.2.x in further subsections for each lncRNA or at least related processes.
Author Response
We thank the reviewer for bringing this to our attention. We have further subdivided some of the sections into subcategories that we think improve the readability of the paper. To do this, we also changed the original order in which some lncRNAs were described, and we made small changes in writing to better fit each lncRNA in its new category.
Reviewer 2 Report
The manuscript "LncRNAs in the type I interferon antiviral response" by Suarez et al is a solid review of current data in the field. The text is organized in classical manner with introduction into the field, so it will be easy to read both by researcher from lncRNA and virology fields. Review covers most of the key studies and at the same time authors do not just summarize the data. There is a flavor of personal viewpoints - so valuable in reviews.
I would suggest to add some discussion of lncRNA roles in antiviral response and development of interferonopathies. There are some similarities and differences that can be of interest for readers.
As a minor point - please move Fig.2 to page 3. This will simplify understanding of complex mechanisms for readers.
I suggest to accept the review after minor revision.
Author Response
We thank the reviewer for his/her positive comments on our work. We have gladly made the changes suggested by the reviewer as we detail below
1. Interferonopathies are indeed a fascinating group of diseases whose study will undoubtedly impact research on the antiviral response and vice versa. In struggling to keep the manuscript from becoming even denser in information, as highlighted by another reviewer, and as suggested by the reviewer, we have now discussed about antiviral response and development of interferonopathies. For clarity, we also included a brief description of interferonopathies. Further, we cite two recent reviews on the subject to help readers interested in the matter (page 21, lines 780-792).
2. We have moved figure 2 to page 3, as suggested by the reviewer.